# *Bacillus aryabhattai* Mitigates the Effects of Salt and Water Stress on the Agronomic Performance of Maize under an Agroecological System

Henderson Castelo Sousa [1,*], Geocleber Gomes de Sousa [2,*], Thales Vinícius de Araújo Viana [1], Arthur Prudêncio de Araújo Pereira [3], Carla Ingryd Nojosa Lessa [1], Maria Vanessa Pires de Souza [1], José Marcelo da Silva Guilherme [1], Geovana Ferreira Goes [1], Francisco Gleyson da Silveira Alves [4], Silas Primola Gomes [2] and Fred Denilson Barbosa da Silva [2]

[1] Agricultural Engineering Department, Federal University of Ceará, Fortaleza 60455-760, Brazil; thales@ufc.br (T.V.d.A.V.); ingrydnojosa@alu.ufc.br (C.I.N.L.); vanessa.pires@alu.ufc.br (M.V.P.d.S.); josemarcelo01@alu.ufc.br (J.M.d.S.G.); geovanagoes@alu.ufc.br (G.F.G.)

[2] Institute of Rural Development, University of International Integration of Afro-Brazilian Lusofonia, Redenção 62790-000, Brazil; silas.primola@unilab.edu.br (S.P.G.); freddenilson@unilab.edu.br (F.D.B.d.S.)

[3] Soil Science Department, Federal University of Ceará, Fortaleza 60355-636, Brazil; arthur.prudencio@ufc.br

[4] Department of Animal Science, Federal University of Ceará, Fortaleza 60356-000, Brazil; gleyson@ufc.br

* Correspondence: henderson@alu.ufc.br (H.C.S.); sousagg@unilab.edu.br (G.G.d.S.)

**Abstract:** The use of plant-growth-promoting rhizobacteria (PGPR) can be one option for mitigating the impact of abiotic constraints on different cropping systems in the tropical semi-arid region. Studies suggest that these bacteria have mechanisms to mitigate the effects of water stress and to promote more significant growth in plant species. These mechanisms involve phenotypic changes in growth, water conservation, plant cell protection, and damage restoration through the integration of phytohormone modulation, stress-induced enzyme apparatus, and metabolites. The aim of this study was to evaluate the growth, leaf gas exchange, and yield in maize (*Zea mays* L.—BRS Caatingueiro) inoculated with *Bacillus aryabhattai* and subjected to water and salt stress. The experiment followed a randomised block design, in a split-plot arrangement, with six repetitions. The plots comprised two levels of electrical conductivity of the irrigation water (0.3 dS m$^{-1}$ and 3.0 dS m$^{-1}$); the subplots consisted of three irrigation depths (50%, 75%, and 100% of the crop evapotranspiration (ETc)); while the sub-subplots included the presence or absence of *B. aryabhattai* inoculant. A water deficit of 50% of the ETc resulted in the principal negative effects on growth, reducing the leaf area and stem diameter. The use of *B. aryabhattai* mitigated salt stress and promoted better leaf gas exchange by increasing the CO$_2$ assimilation rate, stomatal conductance, and internal CO$_2$ concentration. However, irrigation with brackish water (3.0 dS m$^{-1}$) reduced the instantaneous water-use efficiency of the maize. Our results showed that inoculation wiht PGPR mitigates the effect of abiotic stress (salt and water) in maize plants, making it an option in regions with a scarcity of low-salinity water.

**Keywords:** *Zea mays*; abiotic stress; microorganisms; salinity; water deficit

## 1. Introduction

Maize (*Zea mays* L.), with its origin in Central America, is of great economic importance and is cultivated worldwide. In Brazil it is one of the main cereals produced (21,581.9 million hectares), with an emphasis on food for human and animal consumption as well as for bioenergy production [1–4]. The crop has gradually expanded into arid and semi-arid regions, where it helps to solve problems related to food security in places that have limited water resources [5,6]. It is worth noting that maize is considered moderately sensitive to salinity, with a threshold of 1.1 and 1.7 dS m$^{-1}$ for water and soil electrical conductivity, respectively [7].

The semi-arid region of Brazil is considered one of the largest semi-arid regions, with approximately 27 million inhabitants [8], where irrigation is an important tool for ensuring food security [9]. The characteristics of the region are high temperatures, high evapotranspiration, and a low rainfall rate [10,11]. Water shortages and high salt concentrations in the groundwater are problems that limit agricultural production in this region [9,12].

An excess of salts in the soil solution reduces water absorption by plants and alters metabolic and morphological structures, causing a reduction in seed germination, growth, and productivity in agricultural crops [13–15]. Water and salt stress reduce the soil water potential, making the soil solution unavailable, or not readily available, for nutrient uptake by plants. These stresses have a negative effect on physiological processes, causing partial closure of the stomata, limiting the internal $CO_2$ concentration, reducing the rates of photosynthesis and transpiration, and consequently the water-use efficiency and agricultural crop yields worldwide [16–18]. Evaluating the interaction between salt and water stress in the courgette, [19] found a reduction in photosynthesis and transpiration. Similarly, [20] found a reduction in the productivity of peanuts under salt and water stress.

It should be noted that various strategies have been used in the scientific environment to mitigate salt and water stress. One alternative to mitigate the effects of such stress and ensure production in agroecological systems is the use of microbial inoculants formulated with plant-growth-promoting bacteria (PGPB) [21–23]. These microorganisms can offer protection to plants against water deficits by maintaining moisture levels and providing better root development and nutrient supply. Researchers are seeking to identify microorganisms, together with their action mechanisms, that are able to mitigate abiotic stress [24,25]. Various promising studies have found that inoculating maize with beneficial microorganisms results in greater productivity [26].

In this scenario, the use of plant-growth-promoting rhizobacteria (PGPR), especially from the *Bacillus* genus, stands out in plant development. Some of the known mechanisms by which PGPRs can improve plant development include beneficial effects on promoting plant emergence and growth [27], antagonistic activity against phytopathogenic fungi [28], improvement of soil structure (by bacterial exopolysaccharides), provision of N to plants through biological nitrogen fixation, solubilization and mineralization of nutrients, particularly phosphate, and improvement of resistance to non-biological stresses [29]. The strain of *B. aryabhattai* CMAA 1363 was able to provide drought tolerance in maize plants [24].

Given this promising scenario, the present study tested the hypothesis that the use of plant-growth-promoting bacteria mitigates the effect of abiotic stress (salt and water) on the agronomic performance of maize. The aim of this study, therefore, was to evaluate the growth, leaf gas exchange, and production parameters of maize inoculated with *Bacillus aryabhattai* under water and salt stress.

## 2. Material and Methods

### 2.1. Location and Characterisation of the Experimental Area

The experiment was conducted from 25 August to 17 November 2022 (dry season) under field conditions at the Piroás Experimental Farm (PEF) (04°14′53″ S; 38°45′10″ W, at a mean altitude of 240 m), belonging to the Universidade da Integração Internacional da Lusofonia Afro-Brasileira (UNILAB), in Redenção, in the state of Ceará.

The climate in the region is of type BSh' (tropical semi-arid climate), characterized by very hot temperatures, a rainy season during the summer and autumn (February to May), strong insolation, and high evaporation rates [30]. The amount of rainfall and the maximum and minimum air temperature were recorded daily throughout the experiment, as well as the average relative humidity (Figure 1), monitored by means of a data logger (HOBO® U12-012 Temp/RH/Light/Ext).

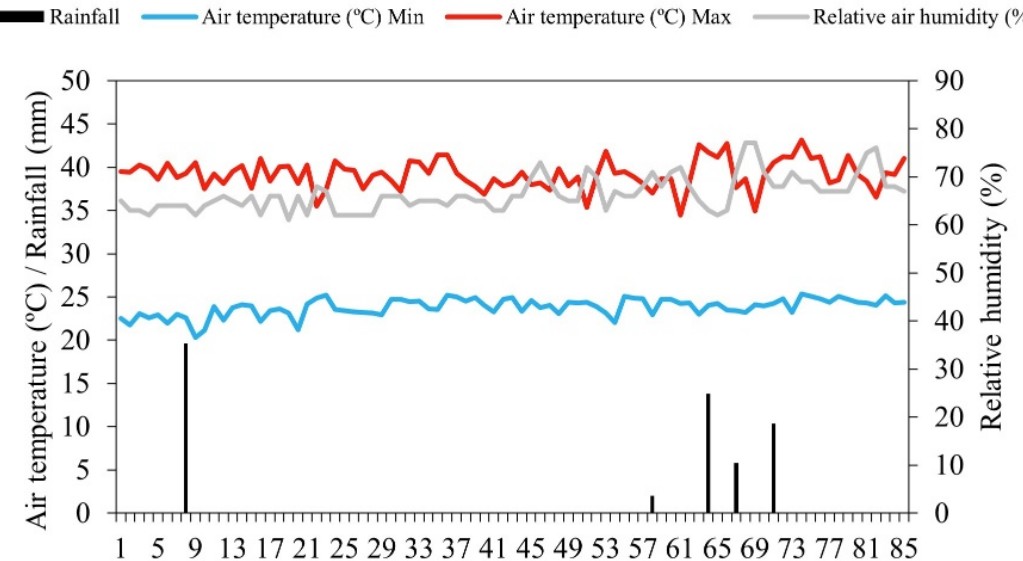

**Figure 1.** Mean values for maximum (Max) and minimum (Min) temperature and relative humidity obtained during the experimental cycle.

The soil in the experimental area is classified as an ultisol. Samples were collected from the surface layer (0–20 cm) and sent to the laboratory to determine the physical and chemical attributes (Table 1), as per the methodology described by [31].

**Table 1.** Chemical and physical characteristics of the soil sample before applying the treatments (0–20 cm).

| pH | OM | N | C | P | Ca | Mg | Na | Al | H + Al | K | ECse | ESP | C/N | V |
|---|---|---|---|---|---|---|---|---|---|---|---|---|---|---|
| H₂O | g kg⁻¹ | | | mg kg⁻¹ | | | cmol_c dm⁻³ | | | | dS m⁻¹ | % | | % |
| 5.6 | 11.59 | 0.71 | 6.72 | 20 | 3.20 | 2.60 | 0.07 | 0.35 | 2.15 | 0.17 | 0.76 | 1 | 9 | 74 |
| SD (g cm⁻³) | | | CS | | FS | | Silt | | Clay | | Textural Classification | | | |
| **Bulk** | **Particle** | | | g kg⁻¹ | | | | | | | | | | |
| 1.31 | 2.61 | | 507 | | 283 | | 133 | | 77 | | Loamy Sand | | | |

OM—Organic matter; ESP—Percentage of exchangeable sodium; ECse—Electrical conductivity of the soil saturation extract; V—Base saturation; SD—Soil density; CS—Coarse sand; FS—Fine sand.

### 2.2. Experimental Design and Treatments

The experimental design was randomised blocks in a split-plot arrangement, with six repetitions. The plots comprised two levels of electrical conductivity of the irrigation water (ECw): water supply (0.3 dS m⁻¹) and a brackish solution (3.0 dS m⁻¹). The sub-plots consisted of three irrigation depths (ID1 = 50%, ID2 = 75%, and ID3 = 100% of the crop evapotranspiration [ETc]). The sub-subplots included the presence or absence of *B. aryabhattai* inoculant (Figure 2).

### 2.3. Irrigation Management

A drip irrigation system was used at a spacing of 0.3 m, corresponding to one emitter per plant. Emitters of 4, 6, and 8 L h⁻¹ were used to standardise the irrigation time, affording water regimes of 50%, 75%, and 100% of the *ETc*, respectively. Uniformity tests were carried out, returning a distribution coefficient of 92%.

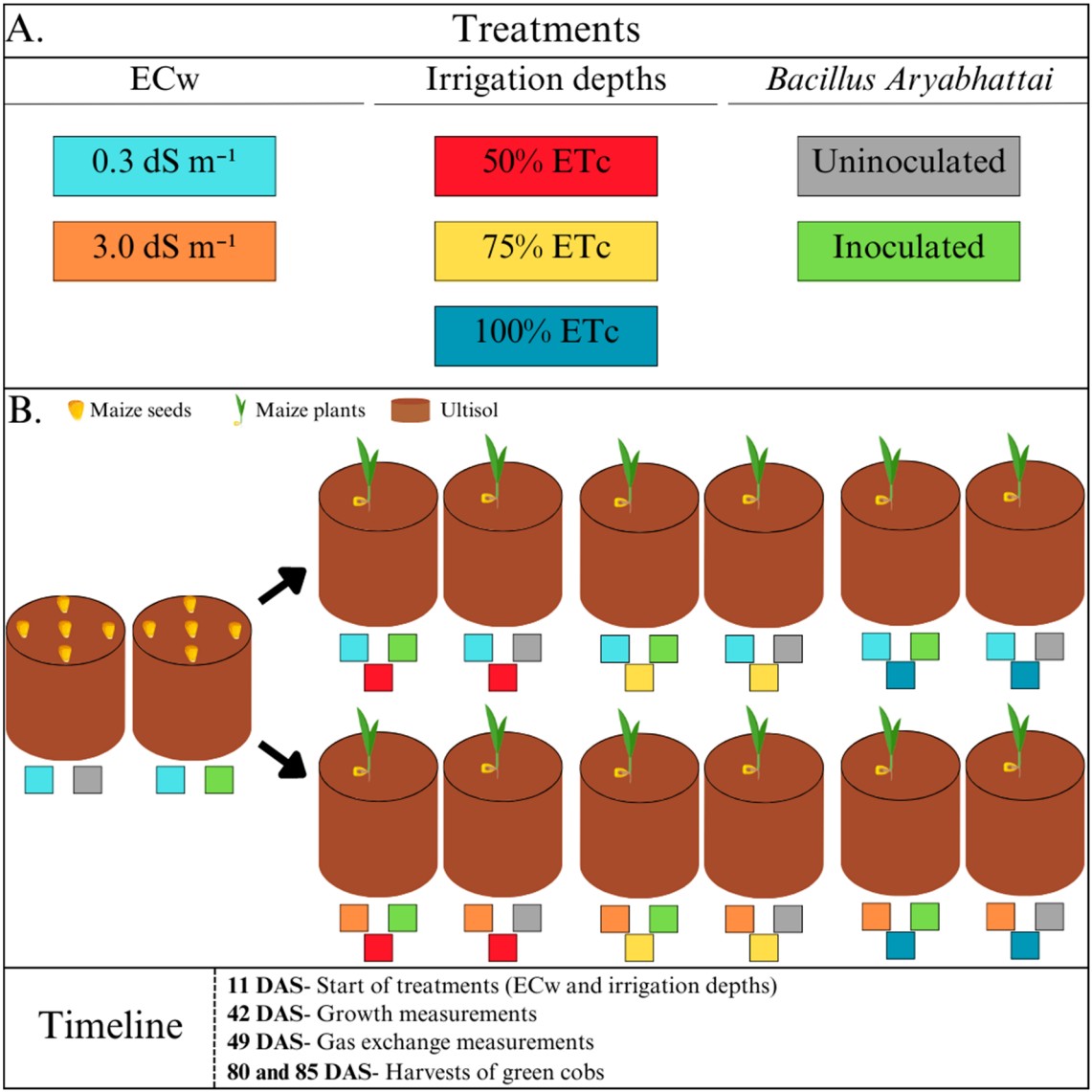

**Figure 2.** Diagram of the experimental design showing (**A**) the composition and interaction of the study factors—electrical conductivity of the water, irrigation depths and inoculation—and (**B**) a timeline of the procedures carried out during the experiment.

Irrigation management was estimated daily from the reference evapotranspiration using data from a Class A evaporimeter pan. The crop evapotranspiration, in mm day$^{-1}$, was calculated from the evaporation measured in the Class A pan, as per Equation (1).

$$ETc = ECA \times Kp \times Kc \tag{1}$$

where:

$ETc$—Crop evapotranspiration, in mm day$^{-1}$;

$ECA$—Evaporation measured in the class A pan, in mm/day$^{-1}$;

$Kp$—Class A pan coefficient, dimensionless;

$Kc$—Crop coefficient, dimensionless.

The following crop coefficients ($Kc$) were adopted: 0.86 (up to 40 days after sowing—DAS); 1.23 (from 41 to 53 DAS); 0.97 (from 54 to 73 DAS), and 0.52 (from 74 DAS to the end

of the cycle) [28]. A leaching fraction of 15% was added to the applied irrigation depth [32]. The irrigation time was obtained using Equation (2):

$$It = \frac{ETc \times Sd}{Af \times q} \times 60 \tag{2}$$

where:

$It$—Irrigation time (min);

$ETc$—Crop evapotranspiration for the period (mm);

$Sd$—Spacing between emitters;

$Af$—Application efficiency (0.92);

$q$—Flow rate (L h$^{-1}$).

Table 2 shows the total irrigation depth applied during the experiment throughout the crop cycle based on each treatment.

**Table 2.** Total irrigation depth applied in each treatment.

| $ECw$ (dS m$^{-1}$) | $ETc$ (%) | Total Depth Applied (mm) | |
|---|---|---|---|
| | | Uninoculated | Inoculated |
| 0.3 | 50 | 260.4 | 260.4 |
| | 75 | 390.6 | 390.6 |
| | 100 | 520.8 | 520.8 |
| 3.0 | 50 | 260.4 | 260.4 |
| | 75 | 390.6 | 390.6 |
| | 100 | 520.8 | 520.8 |

Fresh water (0.3 dS m$^{-1}$) from the dam belonging to FEP was used to irrigate the plants of the control treatment. This same water source was stored in 500 L tanks and used in preparing the 3.0 dS m$^{-1}$ saline solution by dissolving sodium chloride (NaCl), calcium chloride (CaCl$_2$2H$_2$O), and magnesium chloride (MgCl$_2$6H$_2$O), maintaining the proportions predominantly found in the principal water sources of the northeast of Brazil of 7:2:1 [33] and based on the relationship between the ECw and its molar concentration (mmol$_c$ L$^{-1}$ = CE × 10). The electrical conductivity of the water was periodically monitored using a bench conductivity meter (AZ® 806,505 pH/Cond./TDS/Salt). The water was sent for its chemical characteristics to be determined following the methodology of [34] and was classified using the methodology described by [35]. The results are shown in Table 3.

**Table 3.** Chemical characterisation and classification of the irrigation water used in the experiment.

| $ECw$ | $Ca^{2+}$ | $Mg^{2+}$ | $K^+$ | $Na^+$ | $Cl^-$ | $HCO_3^-$ | pH | CE | SAR | Classification [1] |
|---|---|---|---|---|---|---|---|---|---|---|
| dS m$^{-1}$ | | mmol$_c$ L$^{-1}$ | | | mmol L$^{-1}$ | | in H$_2$O | dS m$^{-1}$ | (mmol$_c$ L$^{-1}$)$^{0.5}$ | |
| 0.3 | 0.6 | 1.4 | 0.2 | 0.4 | 2.5 | 0.1 | 6.9 | 0.3 | 0.4 | C$_2$S$_1$ |
| 3.0 | 6.33 | 7.64 | 2.0 | 15.6 | 25 | 1.0 | 7.79 | 3.0 | 5.9 | C$_4$S$_2$ |

[1]—[35]; SAR—Sodium adsorption ratio.

The experiment was irrigated daily with water of 0.3 dS m$^{-1}$ up to 10 days after sowing (DAS) with a water depth of 100% of the ETc. The treatments, including the water regimes and ECw, were started at 11 DAS.

### 2.4. Agroecological Maize Production System (Plant Material, Inoculation, and Fertilisation)

Seeds of the maize (*Zea mays* L.) 'BRS Caatingueiro' variety were used, sown manually with five seeds per hole at a spacing of 0.8 × 0.2 m between the rows and plants. This cultivar is used by producers in the region and has a super-early cycle. At 10 DAS, with the plant stand already established, thinning was carried out to leave one plant per hole.

The inoculation was carried out using the commercial product Auras® (Embrapa and NOOA Agricultural Science and Technology, Patos de Minas–Minas Gerais, Brazil) formulated with *Bacillus aryabhattai* CMAA 1363, licensed by the Brazilian Agricultural Research Corporation (Embrapa, Jaguariúna–São Paulo, Brazil), obtained from the rhizosphere of *Cereus jamaracu*, a cactus present in the Caatinga biome of the Brazilian semi-arid region [36]. The seeds were immersed in the bacterial solution immediately before planting, applying 4 mL kg$^{-1}$ of maize seeds. The rhizobacterium belongs to the inoculant class, with a concentration of $1 \times 10^8$ UFC/mL.

Fertiliser management was based on the chemical analysis of the soil (Table 1) and used organic fertiliser (cattle manure and cattle biofertiliser) applied as a base and topdressing as recommended by [37] for irrigated maize in the state of Ceará, equal to 90 kg ha$^{-1}$ N, 40 kg ha$^{-1}$ P$_2$O$_5$, and 30 kg ha$^{-10}$ K$_2$O.

The chemical characteristics of the cattle manure and cattle biofertiliser were determined as per the methodology of [31] and are shown in Table 4.

**Table 4.** Chemical characterisation of the organic fertilisers used in the experiment.

| Organic Source | N | P | K$^+$ | Ca$^{2+}$ | Mg$^{2+}$ |
|---|---|---|---|---|---|
| | g L$^{-1}$ | | | | |
| Cattle manure | 0.96 | 0.47 | 0.59 | 1.10 | 0.25 |
| Cattle biofertiliser | 0.82 | 1.4 | 1.0 | 2.5 | 0.75 |

*2.5. Variables under Analysis*

2.5.1. Growth

At 42 DAS, the following variables were evaluated: plant height (PH, cm), using a tape, measuring from the soil to the apex of the plant; number of leaves (NL), by directly counting the fully expanded leaves; stem diameter (SD, mm), measured two centimetres from the ground using a pachymeter; leaf area (LA, cm$^2$), using an area integrator (Area meter, LI-3100, Li-Cor, Inc., Lincoln, NE, USA).

2.5.2. Leaf Gas Exchange

At 49 DAS, gas exchange measurements were taken using the third fully expanded leaf from the apex of the plant. The net photosynthetic rate ($A$, μmol CO$_2$ m$^{-2}$ s$^{-1}$), stomatal conductance ($gs$, mol m$^{-2}$ s$^{-1}$), rate of transpiration ($E$, mmol m$^{-2}$ s$^{-1}$), and internal CO$_2$ concentration ($Ci$, μmol mol$^{-1}$) were measured using an infrared gas analyser (Li-6400XT, LICOR, Lincoln, NE, USA) under the following conditions: ambient air temperature, CO$_2$ of 400 ppm, photosynthetically active radiation of 1800 μmol m$^{-2}$ s$^{-1}$, between 09:00 and 11:00. The instantaneous water-use efficiency (WUEi) was estimated from the photosynthesis and transpiration data. The relative chlorophyll index (RCI, SPAD) was measured on the same leaves using a portable meter (SPAD—502 Plus, Minolta, Tokyo, Japan).

2.5.3. Yield

To determine the production parameters, two harvests of green ears were carried out (80 and 85 DAS), when the following were evaluated: ear length (EL, cm), measuring longitudinally using a ruler; ear diameter (ED, mm), measuring transversely using a digital pachymeter; ear yield with straw (EYWS, kg ha$^{-1}$) and ear yield without straw (EYWoS, kg ha$^{-1}$), estimated from the mean weight of the ear and the stipulated plant stand per hectare (62,500 plants ha$^{-1}$).

*2.6. Data Analysis*

The data obtained were subjected to the Kolmogorov–Smirnov test of normality at a level of 0.05 probability. After verifying the normality, analyses of variance were applied using the F-test ($p < 0.05$). In cases of statistical significance, the mean values were compared with Tukey's test ($p < 0.05$) using the Assistat 7.7 Beta software [38].

## 3. Results and Discussion

### 3.1. Growth

The analyses of variance revealed that the leaf area and stalk diameter were significantly influenced by the water regime alone and by the interaction between the electrical conductivity of the water and inoculation. The leaf area was significantly affected by the electrical conductivity of the water and by the interaction between the water regime and inoculation. The triple interaction of the factors ECw × ID × INOC had a significant influence on plant height. The number of leaves was not significantly influenced by any of the factors (Table 5).

**Table 5.** Summary of the analysis of variance for plant height (PH), number of leaves (NL), stem diameter (SD), and leaf area (LA) in maize plants under different levels of electrical conductivity of the irrigation water (*ECw*), irrigation depth (ID), and inoculation (INOC) 42 days after sowing.

| Source of Variation | DF | Mean Square | | | |
|---|---|---|---|---|---|
| | | **PH** | **NL** | **SD** | **LA** |
| Blocks | 5 | 29.67 $^{ns}$ | 1.95 $^{ns}$ | 1.53 $^{ns}$ | 207.38 $^{ns}$ |
| ECw | 1 | 0.06 $^{ns}$ | 2.60 $^{ns}$ | 87.96 ** | 5613.37 * |
| Residual (ECw) | 5 | 27.97 | 0.61 | 2.74 | 412.05 |
| Irrigation depths (ID) | 2 | 21.30 $^{ns}$ | 0.40 $^{ns}$ | 56.08 ** | 10,546.35 ** |
| Residual (ID) | 20 | 15.37 | 0.43 | 2.49 | 974.55 |
| Inoculation (INOC) | 1 | 9.56 $^{ns}$ | 0.33 $^{ns}$ | 35.25 ** | 10,360.73 ** |
| Residual (INOC) | 30 | 16.57 | 0.54 | 3.05 | 1161.23 |
| ECw × ID | 2 | 160.75 ** | 0.25 $^{ns}$ | 1.87 $^{ns}$ | 2864.11 $^{ns}$ |
| ECw × INOC | 1 | 149.91 ** | 0.004 $^{ns}$ | 0.0007 * | 2064.59 $^{ns}$ |
| ID × INOC | 2 | 0.24 * | 0.16 $^{ns}$ | 0.27 $^{ns}$ | 5769.978 * |
| ECw × ID × INOC | 2 | 70.49 * | 1.42 $^{ns}$ | 4.56 $^{ns}$ | 654.36 $^{ns}$ |
| CV (%)—Ecw | | 5.46 | 9.49 | 12.68 | 5.47 |
| CV (%)—ID | | 4.05 | 7.97 | 12.08 | 8.41 |
| CV (%)—INOC | | 4.20 | 8.95 | 13.37 | 9.18 |

DF: Degrees of freedom; CV: Coefficient of variation; $^{ns}$, *, and **: not significant, significant at $p \leq 0.05$, and significant at $p \leq 0.01$, respectively.

The height of the maize plants under low electrical conductivity and full irrigation (100% of the ETc) using water of lower salinity (0.3 dS m$^{-1}$) was greater regardless of inoculation; however, under irrigation at 75% of the ETc, the inoculated plants differed statistically from the uninoculated plants, showing higher values (97.44 cm). Similarly, under irrigation with water of higher salinity (3.0 dS m$^{-1}$), there was a significant difference from the water regime only, of 75%, with the inoculated plants obtaining the highest mean value (100.73 cm) (Figure 3).

The maize plants showed greater height when *B. aryabhattai* was used under a moderate deficit (75% of the ETc), regardless of the quality of the water used, indicating the beneficial effect of this stress condition. Rhizosphere bacteria show beneficial effects in various crops, possessing several mechanisms that help mitigate water stress, especially in relation to strengthening phytohormone activity (abscisic acid, gibberellins, cytokinins, and auxins) [24,39].

The mitigating effect of water stress in maize by bacteria of the genus *Bacillus* was also reported by [40] under the conditions of a reduced water supply (30% of field capacity), where inoculated plants were taller by around 27.29% compared to uninoculated plants. Reference [41] found that the optimal irrigation regime (100%) had a positive influence on the height of maize plants compared to lower percentages (50% and 75%).

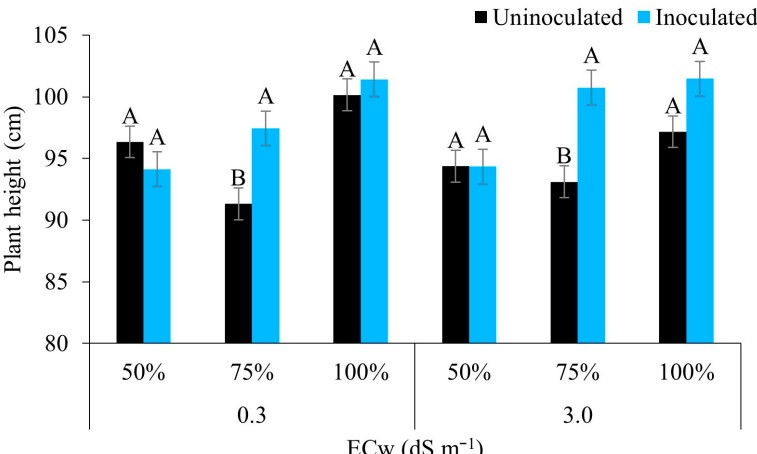

**Figure 3.** Height of maize plants under different levels of electrical conductivity of the irrigation water, different irrigation depths, with and without inoculation, 42 days after sowing. Uppercase letters compare mean values between plants with and without inoculants for the same electrical conductivity and irrigation depth using Tukey's test ($p \leq 0.05$). Error bars represent the standard error of the mean ($n = 6$).

From Figure 4A, it can be seen that between the water regimes, ID1 and ID2, the stem diameter did not differ statistically at the lower values (11.55 and 12.77 mm, respectively), whereas ID3, at 100%, resulted in larger diameters (14.85 mm). Optimal water conditions contribute to turgor pressure, allowing plant cells to develop internal hydrostatic pressure in the cell walls that is essential for cell expansion; on the other hand, a water deficit mainly inhibits leaf expansion and stem growth due to a reduction in pressure [42].

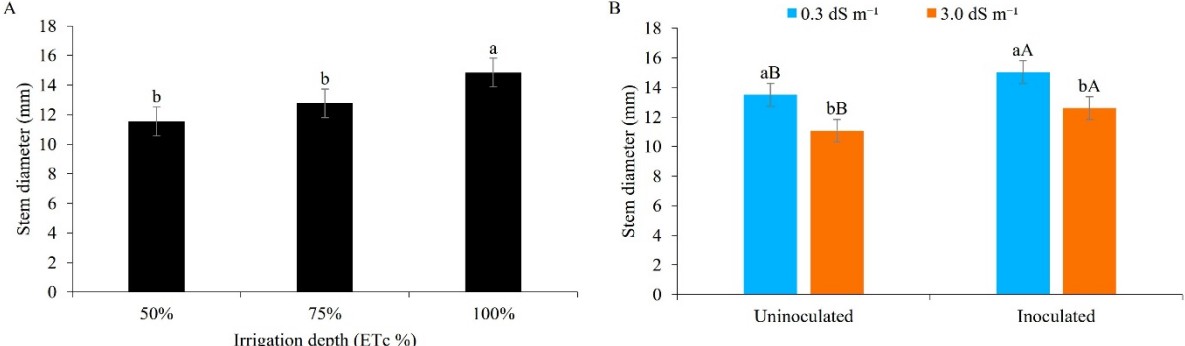

**Figure 4.** Stem diameter of maize plants under different water regimes (**A**) and different levels of electrical conductivity of the irrigation water, with and without inoculation (**B**), 42 days after sowing. (**A**): Lowercase letters compare mean values with Tukey's test ($p \leq 0.05$). (**B**): Lowercase letters compare mean values between ECw levels within each type of inoculation; uppercase letters compare mean values for the type of inoculation within each ECw with Tukey's test ($p \leq 0.05$). Error bars represent the standard error of the mean ($n = 6$).

This result is similar to that of [41], who used different irrigation rates estimated by a class A pan (50%, 75%, 100%, and 125% of the ETc), where the greatest stalk diameter for green maize (11.72 mm) was obtained using the highest rate. Evaluating different irrigation depths in a subsurface drip system, [43] found that reductions starting at 80% of the required depth caused a reduction in the stalk diameter of maize.

The stem diameter was statistically greater when applying water of lower salinity (0.3 dS m$^{-1}$) to inoculated plants, with a mean value of 15.04 mm. When using brackish water, the stem diameter was smaller regardless of inoculation, but showed higher values in inoculated plants in a direct comparison (12.61 mm) (Figure 4B).

The harmful effects of salinity on water and nutrient uptake resulted in a reduction in the stem diameter; however, these effects were mitigated when using *B. aryabhattai*. The presence of rhizobacteria may have mitigated the osmotic effects imposed by salt stress via biochemical changes in the plant or rhizosphere, increasing the physiology of the exposed plants and facilitating water uptake [27,44]. Inoculation with PGPBs during the early stages of maize crop under drought conditions significantly improved the stem diameter [45].

Studying different levels of electrical conductivity for the water (0.2, 1.3, 2.6, 3.9, and 5.2 dS m$^{-1}$), [46] found a linear reduction in the stalk diameter of maize with the increasing salinity of the irrigation water. Similar results were found by [47], who reported that the use of brackish water up to 30 DAS reduced the stem diameter in the cowpea.

It can be seen that the leaf area differed statistically between the levels of electrical conductivity of the irrigation water, with a higher mean value for irrigation water of lower conductivity (0.3 dS m$^{-1}$ = 380.97 cm$^2$) (Figure 5A). The reduction in leaf elongation is a mechanism of survival and water conservation, where under stress conditions, the plants close their stomata and reduce transpiration. In addition, osmotic effects directly interfere with the water uptake of plants [17,48].

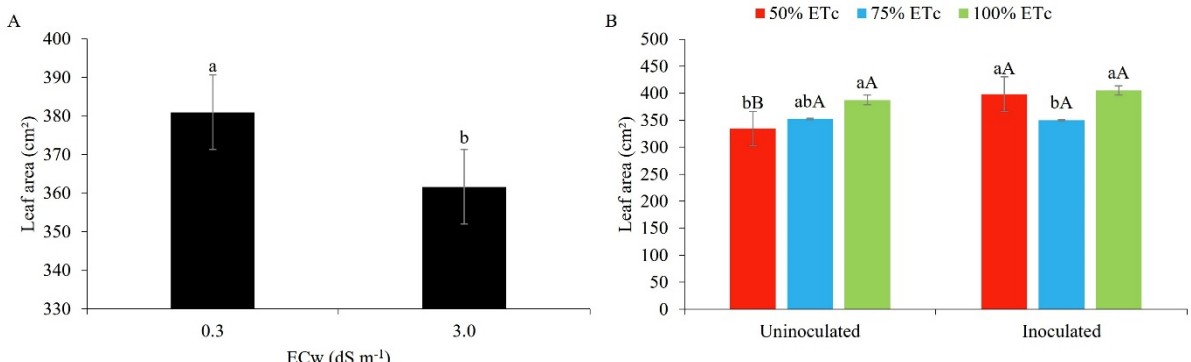

**Figure 5.** Leaf area of maize plants under different levels of electrical conductivity of the irrigation water (**A**) and different irrigation depths, with and without inoculant (**B**), 42 days after sowing. (**A**): Lowercase letters compare mean values using Tukey's test ($p \leq 0.05$). (**B**): Lowercase letters compare mean values between irrigation depths within each type of inoculation; uppercase letters compare mean values for the type of inoculation within each irrigation regime using Tukey's test ($p \leq 0.05$). Error bars represent the standard error of the mean ($n = 6$).

Similar results with maize under salt stress were obtained by [49], where the leaf area underwent a significant reduction of 19.9% in relation to the lowest level of salt (0.5 dS m$^{-1}$). Working with maize, [17] saw a reduction in the leaf area of 15.3% 45 DAS under salt stress (3 dS m$^{-1}$).

As shown in Figure 5B, the leaf area was statistically smaller when associating the water regime of 50% with no inoculant (334.47 cm$^2$); however, in inoculated plants, the ID of 50% (398.19 cm$^2$) and 100% (405.19 cm$^2$) of the Etc were statistically superior to the ID of 75% of the Etc (349.93 cm$^2$).

This result reflects the behaviour of plants subjected to water stress, i.e., they tend to reduce their leaf area as a mechanism for reducing water loss by transpiration, since the water absorption capacity of plants is directly affected by the water content of the soil [5,42]. However, the use of inoculants may have increased colonisation in the soil adhering to the roots, increasing the moisture and improving the ratio of root-adhering soil to root tissue, promoting greater resistance to water stress and consequently, greater leaf area development [25]. Ref. [50], evaluating maize seeds treated with exopolysaccharide-producing bacteria, found an increase in the soil moisture content and a greater leaf area.

### 3.2. Leaf Gas Exchange

As shown in the summary of the analysis of variance of the physiological variables (Table 6), the net photosynthetic rate and the internal $CO_2$ concentration were significantly influenced by the interaction between the electrical conductivity of the water and inoculation. Transpiration, on the other hand, was influenced by the Ecw × ID interaction, while the chlorophyll index was independently influenced by the same factors. The Ecw × ID and ID × INOC interactions influenced the leaf temperature. On the other hand, the electrical conductivity of the water was the single significant factor for water-use efficiency. The triple interaction of the factors Ecw × ID × INOC had a significant influence on stomatal conductance.

**Table 6.** Summary of the analyses of variance for photosynthesis (*A*), stomatal conductance (*gs*), internal $CO_2$ concentration (*Ci*), transpiration (*E*), relative chlorophyll index (RCI), leaf temperature (LT), and instantaneous water-use efficiency (WUEi) in maize plants under different levels of electrical conductivity of the irrigation water (ECw), different irrigation depths (ID), and inoculation (INOC), 49 days after sowing.

| Source of Variation | DF | Mean Square | | | | | | |
| --- | --- | --- | --- | --- | --- | --- | --- | --- |
| | | *A* | *gs* | *Ci* | *E* | RCI | LT | WUEi |
| Blocks | 5 | 9.03 [ns] | 0.67 [ns] | 798.40 [ns] | 0.43 ** | 10.05 [ns] | 3.67 [ns] | 0.006 * |
| ECw | 1 | 361.19 ** | 0.15 [ns] | 4504.68 * | 36.83 ** | 191.12 ** | 74.72 ** | 11.07 ** |
| Residual (ECw) | 5 | 7.25 | 0.16 | 303.63 | 0.00 ** | 2.94 | 0.61 | 0.28 |
| Irrigation depths (ID) | 2 | 2.14 [ns] | 0.26 [ns] | 315.25 [ns] | 0.25 [ns] | 92.35 * | 0.77 * | 0.22 [ns] |
| Residual (ID) | 20 | 8.90 | 0.14 | 101.83 | 0.11 | 23.43 | 0.15 | 0.15 |
| Inoculation (INOC) | 1 | 2.13 [ns] | 3.60 ** | 336.02 [ns] | 0.11 [ns] | 39.45 [ns] | 0.04 * | 0.01 [ns] |
| Residual (INOC) | 30 | 3.13 | 0.18 | 110.47 | 0.16 | 9.78 | 0.02 | 0.21 |
| ECw × ID | 2 | 2.86 [ns] | 1.46 ** | 9.75 [ns] | 0.58 * | 66.17 [ns] | 0.67 * | 0.01 [ns] |
| ECw × INOC | 1 | 6.97 * | 2.48 ** | 595.02 * | 0.47 [ns] | 0.09 [ns] | 0.04 [ns] | 0.008 [ns] |
| ID × INOC | 2 | 1.88 [ns] | 0.04 [ns] | 234.33 [ns] | 0.27 [ns] | 4.97 [ns] | 0.10 * | 0.006 [ns] |
| ECw × ID × INOC | 2 | 0.78 [ns] | 2.57 ** | 110.47 [ns] | 0.18 [ns] | 0.87 [ns] | 0.02 [ns] | 0.27 [ns] |
| CV (%)—ECw | | 11.20 | 11.85 | 6.34 | 0.77 | 5.54 | 2.59 | 9.60 |
| CV (%)—ID | | 12.41 | 10.04 | 3.67 | 7.88 | 15.62 | 1.32 | 7.03 |
| CV (%)—INOC | | 7.37 | 14.05 | 3.82 | 9.11 | 10.09 | 0.50 | 8.31 |

DF: Degrees of freedom; CV: Coefficient of variation; [ns], *, and **: not significant, significant at $p \leq 0.05$, and significant at $p \leq 0.01$, respectively.

From Figure 6, it can be seen that the photosynthetic rate was higher when the maize crop was subjected to irrigation with brackish water (3.0 dS m$^{-1}$), with and without inoculation. This behaviour is possibly linked to the presence of magnesium chloride in the irrigation water of higher salinity, since chloride is a crucial micronutrient in capturing light, helping the function of the enzyme that catalyses the photolysis of water in photosystem II, while magnesium is the central macronutrient of chlorophyll, a molecule located in the chloroplasts, which are responsible for capturing sunlight during photosynthesis [51,52].

When observing the effect of salt stress without the use of inoculants, [26], evaluating the photosynthetic rate of maize in pots under irrigation with brackish water, found a reduction in this variable in the presence of salt stress 45 days after sowing. Reference [53], investigating the effects of water salinity on photosynthesis in peanut plants inoculated with *Bradyrhizobium* sp., found similar results to the present study regarding the mitigating effect of the inoculant in plants grown under salt stress.

Figure 7 shows that in water of lower salinity, the inoculated plants achieved a greater stomatal conductance under the irrigation regimes of 50% and 75%, with no statistical difference for the regime of 100% (full irrigation). At the highest level of salinity, the opposite occurred, where plants with the inoculant achieved a greater stomatal conductance under the irrigation regime of 100% only, the other regimes showing no statistical difference. Reference [54] emphasises that this result may be linked to the participation of resistance-

promoting bacteria, i.e., those that have the ability to branch the roots and release exudates that increase the relative water content of the rhizosphere, thereby better coping with stress conditions.

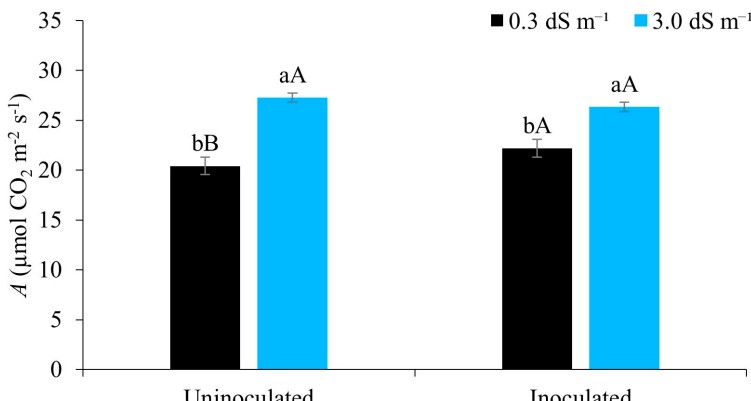

**Figure 6.** Net photosynthetic rate (*A*) in maize plants under different levels of electrical conductivity of the irrigation water, with and without inoculation, 49 days after sowing. Lowercase letters compare mean values between ECw levels within each type of inoculation; uppercase letters compare means values for the type of inoculation within each ECw with Tukey's test ($p \leq 0.05$). Error bars represent the standard error of the mean (*n* = 6).

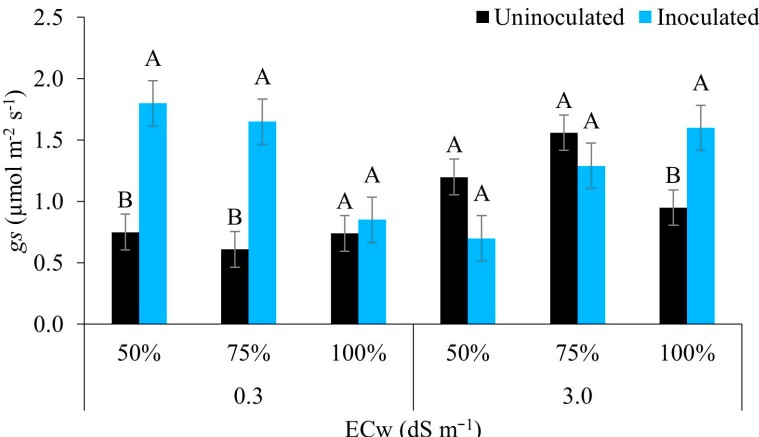

**Figure 7.** Stomatal conductance (*gs*) of maize plants under different levels of electrical conductivity of the irrigation water and different water regimes, with and without inoculation, 49 days after sowing. Uppercase letters compare mean values between plants with and without inoculant within the same electrical conductivity and irrigation depth with Tukey's test ($p \leq 0.05$). Error bars represent the standard error of the mean (*n* = 6).

Studying the courgette irrigated with brackish water under water stress, found that the isolated effect of irrigation water with increasing levels of salts was lower stomatal conductance [19]. However, when using strains of growth-promoting bacteria in maize, [55] reported similar results to the present study. According to those authors, inoculated plants were better able to adjust to stress, showing greater conductance compared to uninoculated plants. Reinforcing the above, ref. [56] described how resistance-promoting bacteria promote a significant increase in osmoprotectants under salt stress, improving the water potential and hydraulic conductivity that positively affect stomatal opening.

It can be seen from Figure 8 that the internal $CO_2$ concentration was higher when the maize was irrigated with water of lower salinity (0.3 dS m$^{-1}$), demonstrating the negative effects of salt stress, which interferes in the osmotic, toxic, and nutritional processes and affects the net $CO_2$ assimilation [18].

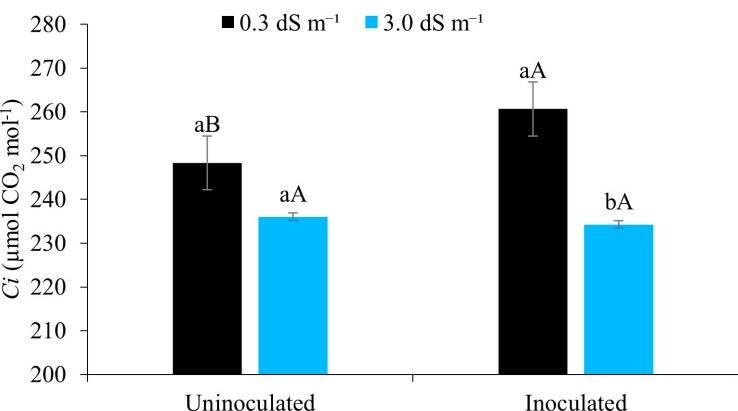

**Figure 8.** Internal $CO_2$ concentration (*Ci*) of maize plants under different levels of electrical conductivity of the irrigation water, with and without inoculation, 49 days after sowing. Lowercase letters compare mean values between ECw levels within each type of inoculation; uppercase letters compare mean values for the type of inoculation within each ECw using Tukey's test ($p \leq 0.05$). Error bars represent the standard error of the mean (*n* = 6).

Similar trends were observed by [57] studying salt stress in okra, where an increase in the electrical conductivity of the irrigation water promoted a reduction in the internal $CO_2$ concentration. The same authors confirm that salt stress induces partial stomatal closure as an attempt by the plant to minimise water loss, which in return reduces the entry of $CO_2$ from the atmosphere into the leaf mesophyll and, since no exchange takes place, reduces its concentration in the substomatal cavity.

According to Figure 9, there was no significant difference in plant transpiration between the water regimes when irrigated with water of lower salinity. However, when compared to higher levels of salinity, the 75% and 100% regimes promoted greater transpiration. Salt and water stress induce osmotic adjustment, which is considered an important mechanism for the maintenance of water uptake and cell turgor under stress conditions [58].

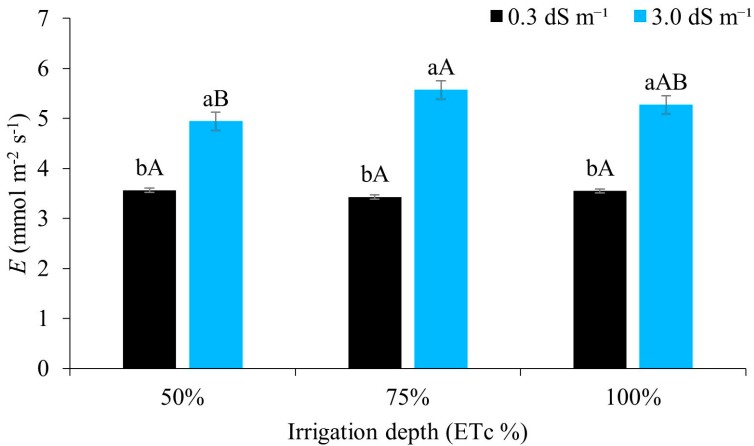

**Figure 9.** Transpiration (*E*) in maize plants under different water regimes with and without inoculant, 49 days after sowing. Lowercase letters compare mean values between ECw levels within each water regime; uppercase letters compare mean values between water regimes at the same ECw with Tukey's test ($p \leq 0.05$). Error bars represent the standard error of the mean (*n* = 6).

Different results, with a reduction in plant transpiration when the electrical conductivity of the irrigation water was increased, were found by [59], cultivating irrigated maize under a water regime of 100% of the ETc. Reference [15] also showed a reduction in transpiration in maize irrigated with brackish water.

The chlorophyll index of the maize was higher under an electrical conductivity of 3.0 dS m$^{-1}$ and was statistically different from the lower salinity (0.3 dS m$^{-1}$) (Figure 10A).

This response may be related to the conditions of low $CO_2$ availability due to stomatal closure and physiological imbalances linked to the high salt content, reaffirming that the chlorophyll content is influenced by both biotic and abiotic factors [42].

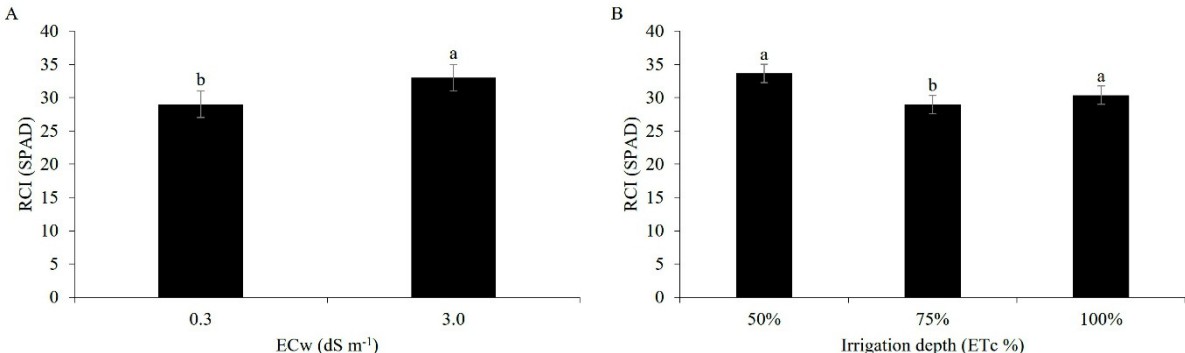

**Figure 10.** Relative chlorophyll index (RCI) of maize plants under irrigation with water of different levels of electrical conductivity (**A**) and different water regimes (**B**), 49 days after sowing. Lowercase letters compare mean values using Tukey's test ($p \leq 0.05$). Error bars represent the standard error of the mean ($n = 6$).

Under the influence of the applied water regimes, the chlorophyll index reached the highest value when 50% of the ETc was used, in relation to the other regimes, showing that under the conditions of the present study the water deficit did not negatively affect this variable (Figure 10B). The opposite result was found in [60], where a reduction in the chlorophyll index followed a reduction in the irrigation depth for five irrigation depths at different sampling times.

The internal leaf temperature (Figure 11A) significantly increased when using water of higher salinity (3.0 dS m$^{-1}$), with an increase of 2.82, 2.64, and 2.04 °C for the irrigation depths of 50%, 75%, and 100%, respectively, compared to water of lower salinity (0.3 dS m$^{-1}$). Following the trend for transpiration under salt stress, the leaf temperature gradually increased. It should be noted that plants under salt stress show great difficulty in absorbing water from the soil; as such, there is an increase in internal temperature, since water helps in the thermal regulation of plants, even under conditions of high transpiration [42–59]. It is worth noting that transpiration via movement of the stomata helps in reducing the leaf temperature (cooling), which is crucial during the day when the leaf absorbs large amounts of energy from the sun [42].

Irrigating peanut plants with brackish water (1, 2, 3, 4, and 5 dS m$^{-1}$), [53] reported a linear increase in the internal leaf temperature. Reference [61], studying maize, also found that salinity afforded an increase in the leaf temperature, reaching 36.7 °C.

It can be seen from Figure 11B that only inoculated plants under the ID of 100% of the ETc differed statistically from the other treatments, with the highest values (30.7 °C). The symbiosis between plants and microorganisms tends to afford better osmotic adjustment, improving transpiration and reducing leaf temperature [62]. Under the conditions of the present study, the heat-dissipation mechanism of the inoculated plants under full irrigation (100%) was possibly not compromised, since the recorded temperatures are within the range for plants with a C4 metabolism, such as maize [42].

Evaluating the peanut under inoculation with *Bradyrhizobium* sp., [53] obtained different results to those of the present study, where inoculated plants showed a lower leaf temperature.

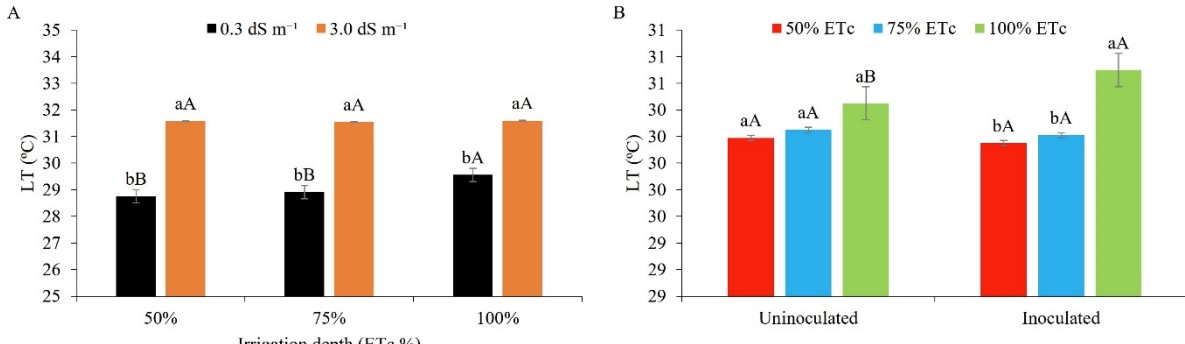

**Figure 11.** Leaf temperature (LT) in maize plants under different levels of electrical conductivity of the irrigation water, different irrigation depths (**A**), and different water regimes, with and without inoculant (**B**), 49 days after sowing. (**A**): Lowercase letters compare mean values between ECw levels within each irrigation depth; uppercase letters compare mean values between irrigation depths at the same ECw with Tukey's test ($p \leq 0.05$). (**B**): Lowercase letters compare mean values between irrigation depths within each type of inoculation; uppercase letters compare mean values between the types of inoculation within each irrigation depth with Tukey's test ($p \leq 0.05$). Error bars represent the standard error of the mean ($n = 6$).

The instantaneous water-use efficiency in maize plants under irrigation at the higher level of salinity (3.0 dS m$^{-1}$) was lower by around 17.6% in relation to irrigation at 0.3 dS m$^{-1}$ (Figure 12). Salt stress induced by irrigation results in limited water uptake due to osmotic and physiological effects, in addition to biochemical changes, which result in a reduced water-use efficiency [15].

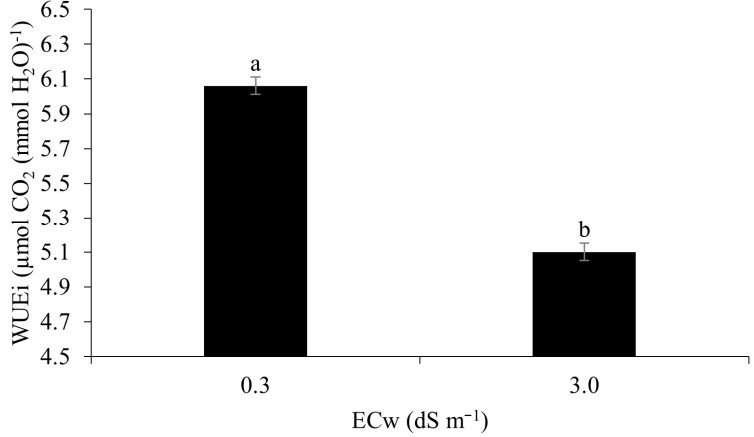

**Figure 12.** Instantaneous water-use efficiency (WUEi) in maize plants under different levels of electrical conductivity of the irrigation water 49 days after sowing. Lowercase letters compare mean values with Tukey's test ($p \leq 0.05$). Error bars represent the standard error of the mean ($n = 6$).

A reduction in the instantaneous water-use efficiency of maize at 49 DAS was also reported by [63] when irrigating the crop with brackish water of 4.5 dS m$^{-1}$ under field conditions in the northeast of Brazil.

*3.3. Yield*

The summary of the analyses of variance of the yield parameters (Table 7) shows the significant influence of the ECw × INOC and ID × INOC interactions on the ear length, while the diameter was not affected by any of the factors. On the other hand, for ears with straw and ears without straw, the yield was significantly influenced by the interaction of the factors under study (ECw × ID × INOC).

**Table 7.** Summary of the analyses of variance for ear length (EL), ear diameter (ED), ear yield with straw (EYWS), and ear yield without straw (EYWoS) in maize plants under different levels of electrical conductivity of the irrigation water (ECw), different irrigation depths (ID), and inoculation (INOC).

| Source of Variation | DF | Mean Square | | | |
|---|---|---|---|---|---|
| | | EL | ED | EYWS | EYWoS |
| Blocks | 5 | 3.09 $^{ns}$ | 17.63 $^{ns}$ | 6,772,473.13 $^{ns}$ | 3,162,178.48 $^{ns}$ |
| ECw | 1 | 5.15 $^{ns}$ | 9.90 $^{ns}$ | 40,862,788.82 ** | 14,079,648.00 ** |
| Residual (ECw) | 5 | 1.69 | 4.58 | 1,711,897.76 | 820,739.41 |
| Irrigation depths (ID) | 2 | 1.35 $^{ns}$ | 5.27 $^{ns}$ | 3,121,961.40 $^{ns}$ | 1,275,678.96 * |
| Residual (ID) | 20 | 1.49 | 3.63 | 1,464,975.55 | 351,695.38 |
| Inoculation (INOC) | 1 | 0.14 $^{ns}$ | 0.38 $^{ns}$ | 3,533,704.78 ** | 450,274.80 * |
| Residual (INOC) | 30 | 0.86 | 1.96 | 363,835.20 | 309,105.44 |
| ECw × ID | 2 | 0.34 $^{ns}$ | 1.57 $^{ns}$ | 5,385,095.95 * | 1,002,466.33 $^{ns}$ |
| ECw × INOC | 1 | 3.78 * | 2.51 $^{ns}$ | 1,257,793.16 $^{ns}$ | 69,497.37 $^{ns}$ |
| ID × INOC | 2 | 6.11 ** | 1.65 $^{ns}$ | 956,622.93 $^{ns}$ | 225,383.24 $^{ns}$ |
| ECw × ID × INOC | 2 | 1.27 $^{ns}$ | 5.31 $^{ns}$ | 1,678,902.68 * | 1,118,325.91 * |
| CV (%)—ECw | | 10.92 | 6.24 | 29.91 | 29.74 |
| CV (%)—ID | | 10.24 | 5.56 | 27.67 | 19.47 |
| CV (%)—INOC | | 7.80 | 4.08 | 13.79 | 18.25 |

DF: Degrees of freedom; CV: Coefficient of variation; $^{ns}$, *, and **: not significant, significant at $p \leq 0.05$, and significant at $p \leq 0.01$, respectively.

The ear length was not statistically affected when using water of lower or higher salinity in the presence of *Bacillus aryabhattai*; however, salt stress promoted greater ear length in inoculated maize plants and was statistically superior to water of lower salinity in the absence of the inoculant (Figure 13A). These data reveal the possible mitigating effect of *Bacillus* for maize in saline environments. The applied PGP bacteria may have generated a mechanism of plant protection against salt stress through the production of auxins and increased nitrogen fixation [21,64]. In addition, they assist the plant in combating physiological drought under salt stress by increasing the water content in the cell [65], reflecting in greater performance for ear length.

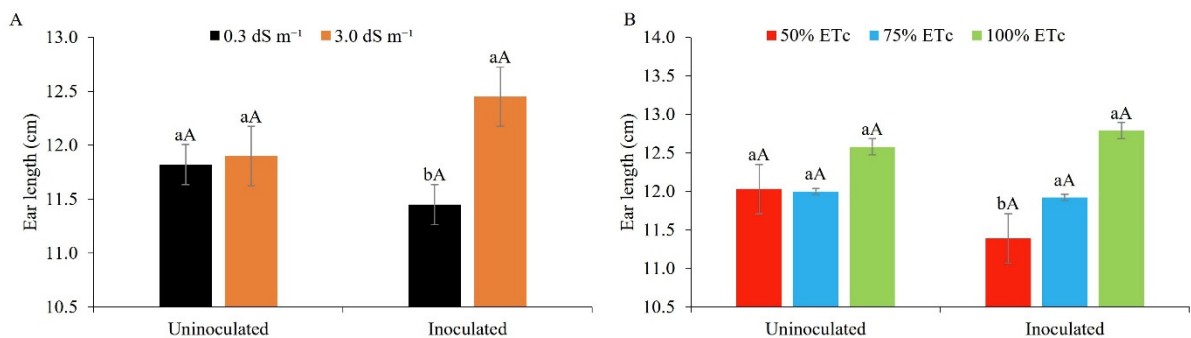

**Figure 13.** Ear length in maize plants under different levels of electrical conductivity of the irrigation water, with and without inoculation (**A**), and different irrigation depths, with and without inoculation (**B**). (**A**): Lowercase letters compare mean values between ECw levels within each type of inoculation; uppercase letters compare mean values for the type of inoculation within each ECw, with Tukey's test ($p \leq 0.05$). (**B**): Lowercase letters compare mean values between irrigation depths within each type of inoculation; uppercase letters compare mean values between the types of inoculation within each water regime with Tukey's test ($p \leq 0.05$). Error bars represent the standard error of the mean ($n = 6$).

Studies using maize seeds inoculated with 'Graminante®', a commercial biotechnological product based on *Azospirillum* spp., and irrigated with low-salinity water returned similar results to the present study for ear length [66]. A reduction in the ear length of maize plants irrigated with brackish water with an electrical conductivity of 3.0 dS m$^{-1}$ under

field conditions was also found by [67]. In this study, the effect of poultry biofertiliser—a mixture of live microorganisms (bacteria, yeasts, algae, and filamentous fungi), which, when available to plants, colonise the rhizosphere and/or the interior of the plant, and promote growth by increasing the supply of primary nutrients—was investigated [68].

Figure 13B shows that the ears of the inoculated plants were superior once irrigated with 75% and 100% of the ETc, differing statistically from the deficit irrigation of 50%, with a superiority of 4.6% and 12.3%, respectively. In the absence of the PGPB, however, there was no difference among the water regimes under study.

The use of *B. Aryabhattai* produces compatible osmolytes, small organic molecules such as betaine that assist during environmental stress [31,69,70], and biofilm formation [71], which acts by forming a hydrated microenvironment around the root, retaining water, and making it available for longer [72], thereby mitigating water stress in maize grown under field conditions in the semi-arid region of the northeast of Brazil. Studies conducted with halotolerant rhizobacteria (HT-PGPR) report that they can also help saline soils to recover their natural balance, promoting benefits for plants grown under saline conditions [73]. The opposite effect to that seen in the present study was reported by [74] when irrigating uninoculated maize with brackish water at 100% of the ETc. The same authors found no significant effect for the ear length.

Irrigation with water of lower salinity at 100% of the ETc in inoculated maize plants (Figure 14A) afforded the highest ear yield with straw (4058 kg ha$^{-1}$), higher than the treatment with 50% and 75% of the ETc. The inoculated plants irrigated with brackish water at 75% of the ETc were statistically superior to those from the other regimes. This effect may be related to the protection imposed on the soil by *B. aryabhattai*, which plays an important role in the rhizosphere, improving the soil structure by increasing the volume of macropores, increasing water availability, and binding cations such as Na$^+$ that help mitigate salt stress [65,71,75]. Growing uninoculated maize under field conditions, irrigated with low-salinity water at 100% of the ETc, recorded superior results to those of the present study, achieving a productivity of 10 t ha$^{-1}$ [76].

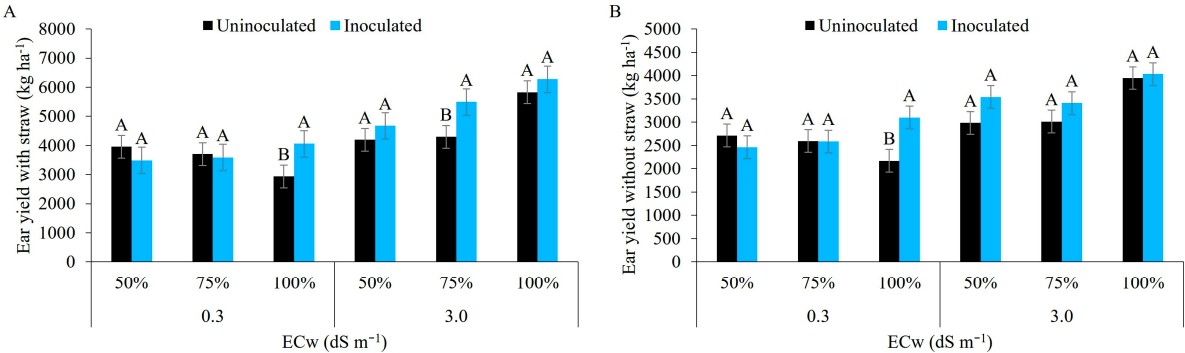

**Figure 14.** Ear yield with straw (**A**) and ear yield without straw (**B**) in maize plants under different levels of electrical conductivity of the irrigation water, different irrigation depths, with and without inoculation. Uppercase letters compare mean values between plants with and without inoculant within the same electrical conductivity and irrigation depth with Tukey's test ($p \leq 0.05$). Error bars represent the standard error of the mean ($n$ = 6).

For the ear yield without straw, the water of lower salinity together with the irrigation depth of 100% of the ETc (3098 kg ha$^{-1}$) was statistically superior to the other treatments, while salt stress showed no statistical difference between the factors under study (Figure 14B). The results obtained in plants under salt and water stresses showed that these stresses, alone or combined, can reduce the productive performance of maize crop. Supporting the findings of this study, [77] describes that the combination of salt and water stress during the reproductive stage can negatively affect the productivity of maize crops. Similar trends to the data found in the present study were reported by [78]

when evaluating the use of *Bacillus subtilis* in seeds of the 'Pioneer 3431' simple hybrid maize cultivar irrigated at 100% of the ETc with low-salinity water. Reference [79], growing uninoculated maize with low-salinity water, found a higher yield than in the present study (6150 kg ha$^{-1}$).

## 4. Conclusions

A water deficit of 50% of the ETc resulted in the principal negative effects on growth, reducing the leaf area and stem diameter. The use of *B. aryabhattai* mitigated salt stress and promoted a better performance in leaf gas exchange by increasing the $CO_2$ assimilation rate, stomatal conductance, and internal $CO_2$ concentration. However, irrigation with brackish water (3.0 dS m$^{-1}$) reduced the instantaneous water-use efficiency of the maize.

Overall, inoculation partially reduced the effects of abiotic stress by means of morphophysiological characteristics, such as increased leaf area and plant height, as well as with no salt stress. These observations reinforce the hypothesis that inoculation mitigates the effect of abiotic stress (salt and water) in maize plants, making it an option in regions with a scarcity of low-salinity water. However, further studies are needed to understand how *B. aryabhattai* acts on morphophysiological and production characteristics under stress conditions in order to develop efficient strategies to mitigate the harmful effects of salt and water stress in the semi-arid region of the northeast of Brazil.

**Author Contributions:** Conceptualization, H.C.S., G.G.d.S., T.V.d.A.V., A.P.d.A.P. and F.D.B.d.S.; methodology, H.C.S., G.G.d.S., T.V.d.A.V., A.P.d.A.P., M.V.P.d.S., F.G.d.S.A. and S.P.G.; investigation, H.C.S., C.I.N.L., M.V.P.d.S., G.G.d.S., G.F.G. and J.M.d.S.G.; writing—original draft preparation, H.C.S., G.G.d.S., C.I.N.L., T.V.d.A.V. and S.P.G.; writing—review and editing, H.C.S., A.P.d.A.P., G.G.d.S., F.G.d.S.A. and F.D.B.d.S.; project administration, G.G.d.S. and T.V.d.A.V. All authors have read and agreed to the published version of the manuscript.

**Funding:** This research received no external funding.

**Institutional Review Board Statement:** Not applicable.

**Data Availability Statement:** Not applicable.

**Acknowledgments:** Acknowledgments are due to the Conselho Nacional de Desenvolvimento Científico e Tecnológico (CNPq), Improvement of Higher Level Personnel Agency (CAPES), for the financial support provided for this research and award of a fellowship to the first author. We would like to thank Itamar Soares de Melo (EMBRAPA Meio Ambiente) for providing the *Bacillus aryabhattai* strains.

**Conflicts of Interest:** The authors declare there to be no conflict of interest.

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
