# Peer review of "Bacillus aryabhattai Mitigates the Effects of Salt and Water Stress on the Agronomic Performance of Maize under an Agroecological System"

_agriculture, doi:10.3390/agriculture13061150_

Round 1

Reviewer 1 Report

The study “Bacillus aryabhattai mitigates the effects of salt and water stress on the agronomic performance of maize under an agro-ecological system” demonstrated that a PGPB strain Bacillus aryabhattai improved the yield and growth of maize under water and salt stresses. The study is lacking important information and I have several concerns regarding the methodology used. Please find the below points and revise the manuscript.

The specie name of the bacterium should not be capitalized in the title

Need to rewrite the abstract, no need to explain in detail about the experimental layout/methods. Just mention the treatments and findings, also write the significance of this study at the end of the abstract

Information about whether the plant cultivar used is susceptible/tolerant to water and salt stress is missing

It is not clear that how the seed application of Bacillus aryabhattai was done? The seeds were dipped into the bacterial suspension or the suspension was sprayed on seeds?

One important concern is that the authors leave out the effect on plant roots. As both water and salt stress firstly and directly affect the plant roots and remaining growth attributes are directly related to the development of plant roots, I think data on root parameters was important

Line 159: the unit (cm) is already mentioned in line 159, no need to repeat it in 160 “measures in centimeters”

Same is for other parameters

Conclusion is a bit lengthy. Delete unnecessary information and try to conclude the findings

There are several recent reports on abiotic stress especially salt stress alleviation effects of plant growth promoting bacteria on different crop plants. Authors should bring in discussion part

Authors should mentioned discuss about the practical application of this proposed treatments especially for economic feasibility

Although mechanism of action is not the part of this study, authors however should explain concisely that how PGPB act to alleviate salt and water stress in plants

The source bacterium is not given

Manuscript should be carefully revised for English language as I found mistakes regarding grammar and sentence structure at several places

Manuscript should be carefully revised for English language as I found mistakes regarding grammar and sentence structure at several places

Author Response

- REVIEWER 1

The study “Bacillus aryabhattai mitigates the effects of salt and water stress on the agronomic performance of maize under an agro-ecological system” demonstrated that a PGPB strain Bacillus aryabhattai improved the yield and growth of maize under water and salt stresses. The study is lacking important information and I have several concerns regarding the methodology used. Please find the below points and revise the manuscript.

- The specie name of the bacterium should not be capitalized in the title

AUTHORS’ ANSWER: The modification has been made to the text.

- Need to rewrite the abstract, no need to explain in detail about the experimental layout/methods. Just mention the treatments and findings, also write the significance of this study at the end of the abstract

AUTHORS’ ANSWER: The suggestion was accepted and the abstract was modified to meet the reviewer's requirements.

 - Information about whether the plant cultivar used is susceptible/tolerant to water and salt stress is missing

AUTHORS’ ANSWER: The information was added to the article text.

 - It is not clear that how the seed application of Bacillus aryabhattai was done? The seeds were dipped into the bacterial suspension or the suspension was sprayed on seeds?

AUTHORS’ ANSWER: The description was detailed for better understanding and to meet the reviewer's request.

 - One important concern is that the authors leave out the effect on plant roots. As both water and salt stress firstly and directly affect the plant roots and remaining growth attributes are directly related to the development of plant roots, I think data on root parameters was important.

AUTHORS’ ANSWER: We understand the importance of root evaluation; however, due to the experiment being conducted in the field and not in pots, the characteristics of the study area did not allow for such evaluations. We believe that the absence of this data does not diminish the relevance of the study.

 - Line 159: the unit (cm) is already mentioned in line 159, no need to repeat it in 160 “measures in centimeters”. Same is for other parameters.

AUTHORS’ ANSWER: We have adjusted the text as requested by the reviewer and removed duplicate information.

 - Conclusion is a bit lengthy. Delete unnecessary information and try to conclude the findings.

AUTHORS’ ANSWER: The conclusion was rewritten according to the reviewer's guidance.

 - There are several recent reports on abiotic stress especially salt stress alleviation effects of plant growth promoting bacteria on different crop plants. Authors should bring in discussion part.

AUTHORS’ ANSWER: New references were added to meet the request throughout the text of the manuscript.

 - Authors should mentioned discuss about the practical application of this proposed treatments especially for economic feasibility.

AUTHORS’ ANSWER: Our study provides practical results focused on agronomic characteristics. For discussing economic feasibility, a study with a specific objective on this matter is necessary since information related to it was not evaluated in the present experiment. We understand that this type of study is extremely important, but it was not the focus of our experiment.

 - Although mechanism of action is not the part of this study, authors however should explain concisely that how PGPB act to alleviate salt and water stress in plants

AUTHORS’ ANSWER: We added information throughout the manuscript to provide a more concise clarification of the effects of PGPB. It is worth noting that the literature on the effects of rhizobacteria on combined abiotic stresses, as in the present study, is scarce.

 - The source bacterium is not given

AUTHORS’ ANSWER: The information was added to the article text.

 - Manuscript should be carefully revised for English language as I found mistakes regarding grammar and sentence structure at several places

AUTHORS’ ANSWER: The review throughout the text was conducted by authors with experience in English-language publications, seeking to meet the reviewer's recommendation.

Reviewer 2 Report

Article No. agriculture-2363346

Article titled Bacillus Aryabhattai mitigates the effects of salt and water stress on the agronomic performance of maize under an agroecological system

Dear authors

The current manuscript has a good objective, and as it is known Plant growth-promoting rhizobacteria (PGPR) can be used to counteract the negative effects of soil salinization on salt-sensitive Maize The use of plant growth-promoting bacteria can be one option for mitigating the impact of abiotic constraints on different cropping systems in the tropical semi-arid region. genus Bacillus aryabhattai is known as PGRP bacteria and tolerant to salinity.

-It needed to be applied in known the mode of action of bacteria as PGRP to complete the objective of the current study

- Abstract:
-it is good, but it short, the authors should consider the proposed changes for improving the clarity of the content. Such add the background on  Bacillus aryabhattai and its effects as PGPR

- It should tell about the full name of the plants for first time (Zea mays L. and what is the source and variety name) and why are you used these plants

-

Keyword: good

-Introduction part is appropriate but a few things are needed for further improvements especially the study aims should be added. Update the references

Need details about the mode of action of PGPR to improve crop productivity and promote plant growth under water and salinity stress

Add some studies about the study highlighting research gaps, which necessitated conducting this trial.

Materials and methods:
-This part describes very well by using suitable subheadings. However, it needs a few modifications and details of how many concentrations of bacteria you used and the tolerance ratio of bacteria to salinity.

-What is the source of bacteria and is it identified by sequencing method or morphology only

- The author said: The climate in the region is type BSh' What is mean of BSh,

- Data logger (HOBO® U12-012 Temp/RH/Light/Ext). add references

- you added the geographic location of the plants you used but need to add the variety of Zea maize  

Salt and temperature tolerance, why are not you testing the strains for temperature tolerance, your study was on halotolerant please clarify.

Results and Discussion
-Both parts need to combine and it needs major revision and it needs some figs of plant treated with and without bacteria under water and salt stress

-Figures b needs draw with a different style to be clearer for readers or add the original picture of experiments.  

- what is the difference between measuring ??? and ?? with and without bacterial treatments  

Plant growth promoting features  it was determined by a preliminary dose/toxicity evaluation of the salt on Z. maize using sodium using chloride (NaCl), calcium chloride (CaCl22H2O) and magnesium     chloride (MgCl26H2O), Where are the results of these  experiments if you do it separately?

- The current study may need gene expression experiment to solve the problem of the mode of action of these bacteria as PGPR bacteria under water and salinity stress

Author said: The present study shows that the use of Bacillus Aryabhattai may have produced compatible osmolytes, small organic molecules such as betaine that assist during environmental stress please add data said that so you said the present study???? And also, biofilm formation, that acts by forming a hydrated microenvironment around the root, retaining water, and making it available for longer

-           

Conclusion:

Authors said: However, this study is the first to report  in a practical way the effect of inoculation with Bacillus aryabhattai on the agronomic performance of maize under salt and water stress in an agroecological system, please documented

References:
-Cross-check the references in the text and reference cite. Few references are not as per journal style in the text as well reference section

-

Article No. agriculture-2363346

Article titled Bacillus Aryabhattai mitigates the effects of salt and water stress on the agronomic performance of maize under an agroecological system

Dear authors

The current manuscript has a good objective, and as it is known Plant growth-promoting rhizobacteria (PGPR) can be used to counteract the negative effects of soil salinization on salt-sensitive Maize The use of plant growth-promoting bacteria can be one option for mitigating the impact of abiotic constraints on different cropping systems in the tropical semi-arid region. genus Bacillus aryabhattai is known as PGRP bacteria and tolerant to salinity.

-It needed to be applied in known the mode of action of bacteria as PGRP to complete the objective of the current study

- Abstract:
-it is good, but it short, the authors should consider the proposed changes for improving the clarity of the content. Such add the background on  Bacillus aryabhattai and its effects as PGPR

- It should tell about the full name of the plants for first time (Zea mays L. and what is the source and variety name) and why are you used these plants

-

Keyword: good

-Introduction part is appropriate but a few things are needed for further improvements especially the study aims should be added. Update the references

Need details about the mode of action of PGPR to improve crop productivity and promote plant growth under water and salinity stress

Add some studies about the study highlighting research gaps, which necessitated conducting this trial.

Materials and methods:
-This part describes very well by using suitable subheadings. However, it needs a few modifications and details of how many concentrations of bacteria you used and the tolerance ratio of bacteria to salinity.

-What is the source of bacteria and is it identified by sequencing method or morphology only

- The author said: The climate in the region is type BSh' What is mean of BSh,

- Data logger (HOBO® U12-012 Temp/RH/Light/Ext). add references

- you added the geographic location of the plants you used but need to add the variety of Zea maize  

Salt and temperature tolerance, why are not you testing the strains for temperature tolerance, your study was on halotolerant please clarify.

Results and Discussion
-Both parts need to combine and it needs major revision and it needs some figs of plant treated with and without bacteria under water and salt stress

-Figures b needs draw with a different style to be clearer for readers or add the original picture of experiments.  

- what is the difference between measuring ??? and ?? with and without bacterial treatments  

Plant growth promoting features  it was determined by a preliminary dose/toxicity evaluation of the salt on Z. maize using sodium using chloride (NaCl), calcium chloride (CaCl22H2O) and magnesium     chloride (MgCl26H2O), Where are the results of these  experiments if you do it separately?

- The current study may need gene expression experiment to solve the problem of the mode of action of these bacteria as PGPR bacteria under water and salinity stress

Author said: The present study shows that the use of Bacillus Aryabhattai may have produced compatible osmolytes, small organic molecules such as betaine that assist during environmental stress please add data said that so you said the present study???? And also, biofilm formation, that acts by forming a hydrated microenvironment around the root, retaining water, and making it available for longer

-           

Conclusion:

Authors said: However, this study is the first to report  in a practical way the effect of inoculation with Bacillus aryabhattai on the agronomic performance of maize under salt and water stress in an agroecological system, please documented

References:
-Cross-check the references in the text and reference cite. Few references are not as per journal style in the text as well reference section

-

Author Response

- REVIEWER 2

Dear authors

The current manuscript has a good objective, and as it is known Plant growth-promoting rhizobacteria (PGPR) can be used to counteract the negative effects of soil salinization on salt-sensitive Maize The use of plant growth-promoting bacteria can be one option for mitigating the impact of abiotic constraints on different cropping systems in the tropical semi-arid region. genus Bacillus aryabhattai is known as PGRP bacteria and tolerant to salinity.

- Abstract:

- It is good, but it short, the authors should consider the proposed changes for improving the clarity of the content. Such add the background on Bacillus aryabhattai and its effects as PGPR.

AUTHORS’ ANSWER: The abstract was improved as suggested.

 - It should tell about the full name of the plants for first time (Zea mays L. and what is the source and variety name) and why are you used these plants.

AUTHORS’ ANSWER: The information was added.

 - Introduction part is appropriate but a few things are needed for further improvements especially the study aims should be added. Update the references.

AUTHORS’ ANSWER: The objective is stated in the last paragraph of the section. We made adjustments throughout the text to meet the reviewer's request.

 - Need details about the mode of action of PGPR to improve crop productivity and promote plant growth under water and salinity stress

AUTHORS’ ANSWER: A paragraph was added in the introduction providing details on the information requested by the reviewer. It is worth noting that studies on the effects of rhizobacteria and combined abiotic stresses, as in the present study, are scarce in the literature.

 - Add some studies about the study highlighting research gaps, which necessitated conducting this trial.

AUTHORS’ ANSWER: They were added in the text.

 - Materials and methods:

- This part describes very well by using suitable subheadings. However, it needs a few modifications and details of how many concentrations of bacteria you used and the tolerance ratio of bacteria to salinity.

AUTHORS’ ANSWER: Thank you for the suggestion. Modifications have been made throughout the section to meet the reviewer's demand.

 - What is the source of bacteria and is it identified by sequencing method or morphology only

AUTHORS’ ANSWER: We provided further information regarding the rhizobacterium used in the text.

 - The author said: The climate in the region is type BSh' What is mean of BSh,

AUTHORS’ ANSWER: The description of the present classification was added to the text to meet the request.

 - You added the geographic location of the plants you used but need to add the variety of Zea maize

AUTHORS’ ANSWER: The information is already presented at the beginning of subsection 2.4, however, we have improved the text to enhance comprehension.

 - Salt and temperature tolerance, why are not you testing the strains for temperature tolerance, your study was on halotolerant please clarify.

AUTHORS’ ANSWER: We understand that tests related to strain temperature tolerance are important for understanding mechanisms related to rhizobacteria. However, our study aimed to evaluate the growth, gas exchange, and production parameters of maize inoculated with Bacillus aryabhattai CMAA 1363 rhizobacteria under abiotic stresses of water and salinity. Therefore, the temperature tolerance of strains is not the focus of this study.

 Results and Discussion

- Both parts need to combine and it needs major revision and it needs some figs of plant treated with and without bacteria under water and salt stress

AUTHORS’ ANSWER: The section has been modified to meet the reviewer's request

 - Figures b needs draw with a different style to be clearer for readers or add the original picture of experiments.

AUTHORS’ ANSWER: The layout of the figures was changed to improve interpretation

 - What is the difference between measuring ??? and ?? with and without bacterial treatments   

AUTHORS’ ANSWER: The cited methodologies are for irrigation estimation. ETc was used to measure irrigation depths based on estimated evapotranspiration, which were applied during the experiment (50%, 75%, and 100%). Meanwhile, it was used to calculate the required irrigation time to reach the depth according to system characteristics.

 - Plant growth promoting features it was determined by a preliminary dose/toxicity evaluation of the salt on Z. maize using sodium using chloride (NaCl), calcium chloride (CaCl22H2O) and magnesium chloride (MgCl26H2O), Where are the results of these experiments if you do it separately?

AUTHORS’ ANSWER: The mentioned methodology was carried out to prepare the brackish water (3.0 dS m-1) used during the irrigation of the experiment, aiming to simulate the composition of brackish waters from Northeast Brazil, according to the methodology described in subitem 2.3 Irrigation management. Therefore, it is not related to a preliminary evaluation. We made an adjustment in the text to improve understanding.  It is worth noting that the salinity studied is above the threshold tolerated by the crop and causes moderate reduction in growth and production.

 - The current study may need gene expression experiment to solve the problem of the mode of action of these bacteria as PGPR bacteria under water and salinity stress.

AUTHORS’ ANSWER: Thank you for the suggestion. We believe that studies with this approach are part of the next steps to understand the mechanisms of action of PGPR in saline environments. However, it is not part of the objective of our experiment. We appreciate the suggestion and we will try to incorporate such analyses in future studies.

 - Author said: The present study shows that the use of Bacillus Aryabhattai may have produced compatible osmolytes, small organic molecules such as betaine that assist during environmental stress please add data said that so you said the present study???? And also, biofilm formation, that acts by forming a hydrated microenvironment around the root, retaining water, and making it available for longer

AUTHORS’ ANSWER: The text has been modified for better understanding and the requested data has been added.

 Conclusion:

- Authors said: However, this study is the first to report in a practical way the effect of inoculation with Bacillus aryabhattai on the agronomic performance of maize under salt and water stress in an agroecological system, please documented.

AUTHORS’ ANSWER: The conclusion was adjusted according to the suggestion of the manuscript reviewer

 References:

- Cross-check the references in the text and reference cite. Few references are not as per journal style in the text as well reference section.

AUTHORS’ ANSWER: The citations and references of this article were revised according to the journal's standards.

Reviewer 3 Report

Dear authors,

please find enclosed a PDF file with specific comments pointing the corrections which should be done.

In the introduction, you are mentioning PGPB in general, but I am missing a brief description on particular species B. aryabhatta (i.e. why did you chose this one for maize inoculation)

Besides, I understand that ECw is familiar to those dealing with amelioration, but could you also provide a brief explanation on the relationship between this and soil salinity since this is ubiquitous in your research.

Please change through all the text the way you cite the literature.

For example: 335

“[18], studying the courgette irrigated with brackish water under water stress, found that...” sounds quite odd.

In particular case it should be:

“Sousa et al. [18], studying the courgette irrigated with brackish water under water stress, found that...”

Kind regards

Author Response

- REVIEWER 3

- In the introduction, you are mentioning PGPB in general, but I am missing a brief description on particular species B. aryabhatta (i.e. why did you chose this one for maize inoculation).

AUTHORS’ ANSWER: The requested information has been added to the text.

 - Besides, I understand that ECw is familiar to those dealing with amelioration, but could you also provide a brief explanation on the relationship between this and soil salinity since this is ubiquitous in your research.

AUTHORS’ ANSWER: The text has been modified according to the proposed changes.

 Please change through all the text the way you cite the literature.

For example: 335

“[18], studying the courgette irrigated with brackish water under water stress, found that...” sounds quite odd.

In particular case it should be:

“Sousa et al. [18], studying the courgette irrigated with brackish water under water stress, found that...”

AUTHORS’ ANSWER: We performed a general review of the citations in this article, in accordance with the journal's guidelines.

Round 2

Reviewer 1 Report

Revisions have been done correctly 

Minor corrections are needed 

Author Response

Following the reviewer's recommendation, an English-language review was carried out.